# Molecular Mechanisms Underlying the Elevated Expression of a Potentially Type 2 Diabetes Mellitus Associated SCD1 Variant

**DOI:** 10.3390/ijms23116221

**Published:** 2022-06-02

**Authors:** Kinga Tibori, Gabriella Orosz, Veronika Zámbó, Péter Szelényi, Farkas Sarnyai, Viola Tamási, Zsolt Rónai, Judit Mátyási, Blanka Tóth, Miklós Csala, Éva Kereszturi

**Affiliations:** 1Department of Molecular Biology, Semmelweis University, H-1085 Budapest, Hungary; tibori.kinga@med.semmelweis-univ.hu (K.T.); orosz.gabriella@med.semmelweis-univ.hu (G.O.); zambo.veronika@med.semmelweis-univ.hu (V.Z.); szelenyi.peter@med.semmelweis-univ.hu (P.S.); sarnyai.farkas@med.semmelweis-univ.hu (F.S.); tamasi.viola@med.semmelweis-univ.hu (V.T.); ronai.zsolt@med.semmelweis-univ.hu (Z.R.); 2Department of Inorganic and Analytical Chemistry, Budapest University of Technology and Economics, H-1111 Budapest, Hungary; matyasi.judit@vbk.bme.hu (J.M.); toth.blanka@vbk.bme.hu (B.T.)

**Keywords:** stearoyl-CoA desaturase-1, single nucleotide polymorphism, fatty acids, type 2 diabetes mellitus, oleate, lipotoxicity, genetic variation, metabolic disorders, obesity

## Abstract

Disturbances in lipid metabolism related to excessive food intake and sedentary lifestyle are among major risk of various metabolic disorders. Stearoyl-CoA desaturase-1 (SCD1) has an essential role in these diseases, as it catalyzes the synthesis of unsaturated fatty acids, both supplying for fat storage and contributing to cellular defense against saturated fatty acid toxicity. Recent studies show that increased activity or over-expression of SCD1 is one of the contributing factors for type 2 diabetes mellitus (T2DM). We aimed to investigate the impact of the common missense rs2234970 (M224L) polymorphism on SCD1 function in transfected cells. We found a higher expression of the minor Leu224 variant, which can be attributed to a combination of mRNA and protein stabilization. The latter was further enhanced by various fatty acids. The increased level of Leu224 variant resulted in an elevated unsaturated: saturated fatty acid ratio, due to higher oleate and palmitoleate contents. Accumulation of Leu224 variant was found in a T2DM patient group, however, the difference was statistically not significant. In conclusion, the minor variant of rs2234970 polymorphism might contribute to the development of obesity-related metabolic disorders, including T2DM, through an increased intracellular level of SCD1.

## 1. Introduction

Stearoyl-CoA desaturase-1 (Δ-9 fatty acid desaturase, SCD1) is localized in the endoplasmic reticulum and it catalyzes the rate-limiting step of unsaturated fatty acid (FA) biosynthesis in human cells by introducing a cis-double bond at Δ9-position in saturated acyl-CoAs. The two major substrates, stearoyl-CoA (18:0) and palmitoyl-CoA (16:0), are converted to oleoyl-CoA (18:1 cisΔ9) and palmitoleoyl-CoA (16:1 cisΔ9), respectively. The enzyme is highly expressed in adipose tissue, liver and lungs; moreover, its gene is significantly induced in liver, skeletal muscle, and heart upon high-carbohydrate diet [1]. The gene possesses an alternative polyadenylation site resulting in two mRNA transcripts, which are supposed to differ in stability and translatability playing a role in rapid and efficient regulation of protein levels [2]. *SCD1* knock-out mice are deficient in triglycerides, cholesteryl esters, and diacylglycerols and were characterized by low plasma levels of oleate and palmitoleate [3]. These animals have a decreased fat mass [4], reduced serum leptin levels [5], and an elevated insulin sensitivity [6]. In turn, increased expression of SCD1 might play a role in the development of the metabolic syndrome [7], which is characterized by central obesity, hypertension, and hypertriglyceridemia in combination with insulin resistance. Metabolic and endocrine dysfunction in this condition can culminate in type 2 diabetes mellitus (T2DM). Insulin resistance demands an increased insulin production, which induces a chronic endoplasmic reticulum stress in the *β*-cells due to a permanently elevated protein synthesis, and it can lead to destruction of the pancreatic islets [8]. Sustained high glucose level results in random, non-enzymatic protein glycation, which contributes to a number of life-threatening consequences, e.g., neuropathy, micro- and macro-angiopathy and, thus, retinopathy, nephropathy, as well as an increased risk of stroke and myocardial infraction. It is important to point out the mutual aggravating effect between T2DM and lipid metabolism leading to a vicious cycle, in which diabetic alterations of lipid metabolism, including an elevated SCD1 level can further enhance the progression of T2DM [9,10].

In addition, increased SCD1 expression and activity have been proved to be implicated in cancer, cardiovascular diseases, and insulin resistance, but their exact molecular mechanism is currently poorly understood [9,10,11,12,13].

The intracellular SCD1 level can vary rapidly due to the short half-life of the protein. The polypeptide contains a PEST degradation domain and may be degraded by cytosolic proteases, presumably by the proteasome in the ERAD process. Stabilization of SCD1 through tyrosine phosphorylation has been demonstrated in lung cancer cells [14], however, a FA-dependent control of protein half-life, i.e., an unsaturated FA-induced degradation, has been described only in Drosophila, but not for the human ortholog [15].

Transcription of *SCD1* is regulated by numerous transcription factors and regions, which are highly conserved among species (e.g., LXR, SREBP, PPAR, PUFARE, C/EBP, TR, and LepRE). Several hormones and nutrients have been shown to activate (insulin, growth factors, glucose, sucrose, and cholesterol) or inhibit (leptin, glucagon, docosahexaenoic acid, and arachidonic acid) the expression of *SCD1* [1,16]. As it might be expected, *SCD1* mRNA level is increased by saturated FAs, e.g., palmitate or stearate, while it is decreased by cis unsaturated FAs, e.g., oleate; however, the latter one is a mild effect only [17]. In addition, cis polyunsaturated FAs (linoleate or linolenate) can also slightly modulate the intracellular *SCD1* mRNA pool [18].

A missense polymorphism (rs2234970, A/C, M224L) of SCD1 has been detected with high occurrence (minor allele frequency: 40%), and it has been investigated in association studies regarding metabolic and cardiovascular diseases as well as different cancer types with conflicting results so far. Most importantly, the Leu/Leu genotype was shown to be associated with higher unsaturated: saturated ratio of C:18 FAs in a healthy population, and it has been suggested that this single nucleotide polymorphism (SNP) may influence SCD activity through the amino acid replacement [19]. Although, rs2234970 polymorphism appears to be associated with lipid metabolism-related conditions, the molecular background is currently unclear.

In the present study, we examined the impact of rs2234970 polymorphism on intracellular SCD1 level and its regulation. We found that the Leu224 variant results in higher intracellular enzyme levels due to a combination of (*i*) increased mRNA stability, (*ii*) decreased protein degradation rate, and (*iii*) susceptibility to protein stabilizing effects of different FAs. Moreover, we found that expression of the Leu224 variant increases the cellular unsaturated: saturated FA ratio more effectively than that of the Met224 version of the enzyme. A mild, yet statistically not significant, accumulation of the Leu224 allele was also found in a group of T2DM patients, supporting both the medical relevance of this polymorphism and the complexity of molecular mechanisms in the development of metabolic disorders.

## 2. Results

### 2.1. Effect of a Single Missense Polymorphism on Intracellular Protein Levels of Human Stearoyl-CoA Desaturase-1 (SCD1)

HEK293T cells were transiently transfected to express the A (Met224) and C (Leu224) M224L human SCD1 variants. 24 h after transfection, cells were harvested and SCD1 protein content was analyzed. No endogenous SCD1 protein was detected in non-transfected and empty vector transfected control cells, but a massive over-expression of the Met224 protein was revealed by immunoblot (Figure 1a).

Significantly higher intracellular protein levels were detected for the Leu224 polymorphic variation of SCD1 compared to the major Met224 protein (Figure 1a,c). To rule out the possibility that increased immunoblot densities are due to an increased affinity of the antibody, Leu224 and Met224 SCD1 versions were also expressed with a C terminal Glu-Glu epitope tag [20] in HEK293T cells. The amount of the tagged proteins showed the same pattern (Figure 1b,c) as the corresponding untagged versions (Figure 1a,c) confirming an approximately 2-times higher level of the Leu224 form. To test the potential cell specificity of the phenomenon, the experiments were repeated by using another cell line. Transfected HepG2 cells also contained notably more Leu224 than Met224 SCD1 protein. Expression of either the tagged (Appendix A) or the untagged (Appendix A) versions resulted in a similar, statistically significant two-fold difference in favor of the minor Leu224 variant (Appendix A).

The observed effect of M224L polymorphism on intracellular SCD1 protein level may be attributed to one or both of two major mechanisms, i.e., elevated mRNA levels allowing enhanced activity of translation or/and increased protein stability. To address this question, mRNA levels, as well as intracellular degradation of the Met224 and Leu224 forms were analyzed.

#### 2.1.1. mRNA Stability of M224L Variants of Stearoyl-CoA Desaturase-1

Total RNA was isolated from transiently transfected HEK293T cells, and genomic and plasmid DNA-free samples were reverse transcribed into cDNA. A and C allele containing *SCD1* mRNA levels were compared by using qPCR, and *GAPDH* served as an endogenous control. The primers designed for *SCD1* amplification did not differentiate between endogenous and over-expressed untagged *SCD1* mRNA in the quantitative assay, whereas the antisense oligo designed for the Glu-Glu tag sequence allowed a specific detection of the mRNA derived from the recombinant expression construct. In either case, however, the C (Leu224) version yielded significantly, 1.889-fold (untagged) and 1.972-fold (tagged) higher mRNA levels (Figure 2a).

Since the mRNA molecules of A (Met224) and C (Leu224) desaturase are otherwise identical (i.e., they possess the same 5′ UTR, 3′ UTR regions, and polyA tail) in vivo as well as in vitro, an alteration of mRNA stability caused by the polymorphism itself seems to be the only possible reason for the difference between their levels. Therefore, we analyzed the mRNA sequences of A and C *SCD1* by using the RNAfold web-based application. The minimum free energy (MFE) structure obtained by this in silico analysis revealed a significant rearrangement in the central segment adjacent to the SNP, i.e., an extra hairpin loop was formed in case of the variant C (Figure 2b). The mountain plot representation of the MFE structure also confirmed that an extra structural element may appear in the flanking region of the polymorphic site, which may contribute to the enhanced stability and more efficient translation of C allele containing mRNA (Figure 2c) [21].

The change in allele-specific mRNA stability proposed on the basis of in silico analysis was also confirmed by in vitro experiments. Intracellular degradation of A and C variants of *SCD1* mRNA was compared using actinomycin D as a transcription inhibitor. HEK293T cells were transiently transfected with Glu-Glu tagged version of the two polymorphic variants, and 24 h later, the cells were treated by 5 µg/mL actinomycin D for 0, 1, 2, 4, 8, or 12 h and harvested for qPCR analysis. The DNA contamination-free total RNA samples were reverse transcribed into cDNA and analyzed by Glu-Glu tag specific oligos to differentiate them from endogenous *SCD1* expression. The stable transcript of *GAPDH* served as endogenous control. A significant difference between polymorphic variants A and C was revealed as soon as 1 h after transcription arrest (Figure 2d). The *SCD1* transcript harboring allele A was reduced by 38% of the initial amount, while the decrement in variant C was only 14.5%. The difference widened further to only 34% of the A allele mRNA and more than 80% of the C allele transcript present at 4 h after transcription arrest in the cells. This difference was essentially maintained until the end of the experiment (Figure 2d).

#### 2.1.2. Intracellular Degradation of M224L Variants of Stearoyl-CoA Desaturase-1

The reportedly short half-life of SCD1 protein [22] offers impeded proteolysis as a potential alternative molecular mechanism besides the increased mRNA stability underlying the high levels of Leu224 variant. Intracellular degradation of Met224 and Leu224 SCD1 molecules was compared using cycloheximide, a cell-permeable inhibitor of translation. Transiently transfected HEK293T cells were harvested for SCD1 immunoblot analysis 0, 1, 2, 4, and 6 h after translation arrest. The amount of Met224 SCD1 protein decreased progressively during the 6-h-long cycloheximide treatment, and its initial value got nearly halved within the first hour after translation arrest. The Leu224 version was found to be significantly more stable, the amount of this variant was still 79% of the initial at the end of the second hour when the Met224 protein was reduced to 36% (Figure 3a). The significant difference in the rate of degradation persisted throughout the 6-h-long treatment.

Met224 is located in the third transmembrane region of the desaturase enzyme. According to I-TASSER prediction software, substitution of Met224 to Leu (M224L) does not lead to any substantial change in the protein structure (Figure 3b). However, an alteration in the normalized B-factor values in this section of the protein indicates a decreased conformational entropy of Leucin residue at position 224 (Figure 3c).

### 2.2. Fatty Acid Induced Protein Stabilization of Leu224 SCD1 Variant

The modulating effect of certain FAs on the level of intracellular SCD1 protein has been demonstrated earlier [23,24]. To test if M224L variants are affected differently by various saturated and unsaturated FAs, transiently transfected HEK293T cells were treated with BSA-conjugated oleate, palmitate, palmitoleate, linoleate, and stearate at a final concentration of 100 µM for 6 h and their SCD1 content was assessed by immunoblotting. The amount of the Met224 protein dropped to less than its half upon oleate in agreement with earlier results, however, the difference was statistically not significant (Figure 4a,c).

Other FAs did not affect the intracellular amount of this major variant (Figure 4a,c). In contrast, almost all the investigated FAs caused a remarkable elevation in the Leu224 enzyme level compared to either the untreated control or the levels of Met224 protein under the corresponding treatment (Figure 4b). The most significant effects were observed for palmitoleate, linoleate, and stearate, resulting in as large as 3-fold, 4-fold, and 6-fold increase in Leu224 SCD1 levels, respectively, compared to the untreated control (Figure 4c). Since none of the FAs altered the mRNA levels of either the A or the C *SCD1* variant during this short-term treatment (Appendix A), this effect can be attributed clearly to translational or post-translational events.

### 2.3. Intensity of Δ9 Desaturation in Transiently Transfected HEK293T Cells

FA content of the cells was analyzed by GC-FID, and SCD1-dependent desaturation was characterized by the calculated ratio between the sum of two major monounsaturated and the sum of two major saturated FAs, i.e., (palmitoleate (C16:1 cisΔ9) + oleate (C18:1 cisΔ9)): (palmitate (C16:0) + stearate (C18:0)). Transfection with an empty vector did not change the cellular FA composition (Figure 5 and Appendix A).

As expected, transfection with a Met224 SCD1 expression vector led to a significant increase in the unsaturated: saturated FA ratio compared to control (1.43 ± 0.02 vs. 1.19 ± 0.01, Figure 5a). In accordance with the higher enzyme levels, the cells expressing the Leu224 variant showed a significantly elevated unsaturated: saturated FA ratio, indicating a more intense intracellular desaturation (1.69 ± 0.01, Figure 5a). In line with this, the absolute amount of the two main products of SCD1, i.e., palmitoleate and oleate, was significantly higher (1.565 times; *p* < 0.05 and 1.867 times; *p* < 0.01, respectively) in the Leu224 SCD1 expressing cells compared to those transfected with the Met224 enzyme (Figure 5b,c).

### 2.4. Effect of Met→Ala Replacement at Position 224 on the Expression and Stability of SCD1

An artificial SCD1 variant possessing an M224A amino acid replacement was generated to reveal how the absence of methionine 224 and the presence of leucine 224 contribute to the observed stabilization of the Leu224 protein. Site-directed mutagenesis was used to change the methionine at position 224 to an alanine in both the unlabeled and the tagged SCD1 expression constructs. The protein level of Ala224 variant was not significantly different from that of the original major Met224 version of the enzyme in HEK293T cells transiently transfected with either the untagged (Figure 6a,c) or tagged (Figure 6b,c) construct.

Time course of the intracellular degradation of Ala224 protein was assessed by applying a cycloheximide treatment. The results showing a slower proteolysis of Ala224 variant (Figure 7a) were comparable to those obtained with the Leu224 protein.

### 2.5. Association of M224L SCD1 Polymorphism with Type 2 Diabetes Mellitus

Possible association between rs2234970 A/C (M224L) SNP and T2DM was assessed by case–control setup. Results are summarized in Table 1.

The observed genotype distribution was exactly as expected on the basis of Hardy–Weinberg equilibrium in the control group (*χ*^2^-test *p* = 0.911), and allele frequencies were concordant with the data available in GnomAD. Association analysis was carried out by using both allele- and genotype-wise approaches including the dominant model (i.e., Genotype combination). As the table shows, frequency of the allele coding for leucine was slightly, but not significantly, higher in the T2DM patient group in both comparisons.

## 3. Discussion

Metabolic disorders, such as obesity, the metabolic syndrome, and T2DM, as well as the tightly related cardiovascular diseases are complex multifactorial conditions, which develop on the basis of multiple genetic and environmental factors. Lipid metabolism is undoubtedly highly relevant in these regards, and the genetic and environmental factors influencing the ratio of saturated and unsaturated FAs are of key importance. While these environmental factors, basically the composition of dietary lipids, i.e., the advantage of vegetable oils rich in unsaturated, particularly n-3 polyunsaturated FAs vs. animal fats containing more saturated FAs, are well established and widely accepted, the genetic factors remain to be elucidated.

Human de novo FA synthesis yields palmitate, which can be elongated with two-carbon units to longer saturated chains, thus, the balanced production of saturated and unsaturated endogenous FAs, which is a prerequisite for the maintenance of optimal membrane fluidity and for an efficient triglyceride synthesis relies on the functioning of desaturation. Although double bonds can be formed at various positions up to 9, the first one must be created by a stearoyl-CoA desaturase between the carbons 9 and 10, which makes SCD activity rate determining for the overall desaturation process. It has also been demonstrated that SCD activity is not limited by the capacity of the associated electron transfer chain, so it is determined by the level of SCD enzyme itself [25]. Among the two human SCDs, SCD1 is highly expressed in most tissues including liver, adipose tissue, heart, lung, and the central nervous system, and its metabolic role is clear, the recently discovered SCD5 is present mostly in the central nervous system and some endocrine and reproductive organs, and its function is obscure [26].

The potential health impact of the expression level and activity of SCD1 can be considered from two major aspects. On the one hand, desaturation provides the cells with a defense mechanism against saturated FA induced lipotoxicity that plays an important role in metabolic and cardiovascular diseases [27,28]. On the other hand, the fat storage efficiency of the body as a whole largely depends on the desaturation capacity of adipocytes and hepatocytes, so an elevated SCD1 expression might increase the risk of obesity and obesity-related diseases.

Adipocyte hypertrophy and a local inflammation in the adipose tissue might lead to a permanent FA release in obesity. The excessive FA supply overloads the oxidative capacity of the cells leading to oxidative stress, endoplasmic reticulum stress, abnormal lipid accumulation, disturbance of cell functions, or even programmed cell death in various tissues (e.g., liver, muscle, heart, and pancreatic islets) [27,28]. These cell-damaging effects of FAs are usually referred to as lipotoxicity, and their role in the development of obesity-related diseases, including the metabolic syndrome and T2DM, receives a growing attention [29,30]. In vitro studies, in line with in vivo observations, repeatedly demonstrated that saturated FA supplementation alone is far more damaging than the administration of unsaturated FAs or a balanced combination of saturated and unsaturated FAs. For instance, the saturated FA palmitate (C16:0) is highly toxic to cultured cells, and its toxicity can be ameliorated by a simultaneous addition of the less deleterious oleate (C18:1 cis9) [31,32,33]. It is, therefore, not surprising that SCD1 induction or over-expression reduces [34,35,36,37,38] while SCD1 inhibition or silencing aggravates [35,38,39,40] palmitate-induced cell damages in vitro, and the protective role of SCD1 is also supported by the phenotype of transgenic mice over-expressing the enzyme selectively in skeletal muscle [41].

Though SCD1 expression and activity can significantly influence both the predisposition to obesity and the cellular vulnerability to lipotoxicity, association studies regarding SCD1 polymorphism and human diseases are scarce. Among the missense polymorphisms of SCD1, only M224L (rs2234970) has been reported to have a relatively high frequency in all studied populations to date (Leu224 allele: 24–53%). Association of this SNP with pathological conditions has been investigated only in four studies so far.

The plasma triglyceride lowering effect of dietary docosahexaenoic acid did not correlate with the polymorphism in subjects with metabolic syndrome [42]. The major Met224 allele was found to be associated with an increased accumulation of intramyocellular lipids in an Indian population, and this condition is known to be linked to skeletal muscle insulin resistance and to the risk of T2DM [43]. The Leu/Leu genotype, however, was shown to associate with higher unsaturated: saturated ratio of C:18 fatty acids in a healthy population both before and after n-3 polyunsaturated FA supplementation, and this metabolic profile is indicative of an increased cardiovascular risk [19]. In addition, the minor Leu224 allele was found to significantly worsen the clinical outcome in patients with stage II colorectal cancer [44]. Taking these together, rs2234970 SNP appears to be associated with a variety of medical conditions, but in the lack of functional studies, the molecular background of these effects is currently unclear. This is why we aimed to elucidate how this polymorphism affects mRNA and protein levels and activity of SCD1 in a cellular model.

Our experiments revealed markedly higher expression and activity of the Leu224 variant, and a more detailed investigation showed a combined molecular mechanism in the background of the observed phenomenon.

Considering the identical promoters and untranslated regions of the two forms, it seems plausible that the higher mRNA level of the Leu224 (C) version can be attributed to increased mRNA stability. In general, polymorphisms in the 3‘UTR as well as other variants (e.g., length, regulation by miRNAs, etc.) affect mRNA degradation [45], and there is also evidence in the literature that common variants of the coding region, even synonymous SNPs, may also alter mRNA stability [46]. Our assumption is substantiated by an extra hairpin loop predicted in the flanking region of the polymorphic site in the minor variant in silico (Figure 2b,c) and confirmed by a hindered mRNA degradation seen after actinomycin D treatment in vitro (Figure 2d).

Our results also demonstrated that the effect of the elevated mRNA level was enhanced by a slower intracellular protein degradation of the Leu224 variant (Figure 3). Alteration of protein stability is known to be related to the primary structure of the polypeptide chain. Missense polymorphisms causing different amino acid changes in various locations have been demonstrated to influence the rate of protein degradation [47,48,49,50].

Interestingly, the increased stability of Leu224 variant could be observed at a third level, too, as the intracellular amount of the protein was further elevated by supplementation with certain FAs (Figure 4). This phenomenon developed solely by protein stabilization as no change in the mRNA levels could be detected. Understanding the stabilizing effect of certain FAs is of special practical importance, as certain therapeutic molecules were shown to have more favorable pharmacokinetic parameters and enhanced stability upon conjugation with FAs [51].

To further elucidate the molecular mechanism underlying these effects, we aimed to investigate whether they can be attributed to the lack of Leu or to the appearance of Met at the polymorphic locus. Analysis of artificial mutations is applicable to address this question [52], as introduction of an alanine to the investigated position can serve as an indifferent reference in the functional assays [53]. Using this approach, it was demonstrated that the increased mRNA stability is caused by the appearance of C in the Leu-coding allele, as the artificial Ala variant did not raise mRNA levels. On the other hand, the protein stabilizing effect was associated with the elimination of methionine side chain, since a similar effect could be detected in case of the artificial Ala224 variant (Figure 6 and Figure 7). In fact, the comparable intracellular expression of the major natural Met224 and artificial Ala224 protein variants seems to be an overall outcome of two opposing effects, i.e., reduction of the mRNA level combined with extension of protein half-life in the Ala224 version. The protein stabilizing effect of certain FAs could again be observed solely in case of the Leu224 variant. It can be hypothesized that the Leu224 form of decreased conformational entropy in the third transmembrane region provides an enhanced sensitivity to various fatty acyl chains.

In summary, our results suggest that rs2234970 polymorphism affects SCD1 protein levels in multiple ways. The C base in the code for Leu leads to an enhanced stability of the mRNA, the presence of Leu224 allows a FA-induced protein stabilization, whereas the loss of methionine at the same position results in a hindered protein degradation (Figure 8).

Our findings provide a mechanistic explanation to the observation that people with the Leu/Leu genotype show higher unsaturated: saturated FA ratio [19], which might increase the risk of obesity-related diseases. Elevated expression of SCD1 is related to T2DM [9,10], and the enzyme might be a promising target in the treatment of the disease, and efforts have been made to develop liver-targeted inhibitors to decrease SCD1 activity [54]. It is notable, however, that polymorphic variants can influence the efficiency of therapy [55,56], thus, SCD1 polymorphisms including M224L SNP might alter the effect or even the need of such SCD1-inhibitors. This is another consideration supporting that the development of individualized therapeutic protocols based on genetic profiling would be the future objective in the treatment of T2DM, similarly to other diseases [55]. This goal, however, can be achieved only if the pathomechanism is understood at molecular level, which necessitates a detailed functional characterization of polymorphisms in genes associated to the disease.

## 4. Materials and Methods

### 4.1. Chemicals and Materials

Culture medium and supplements were purchased from Thermo Fisher Scientific (Waltham, MA, USA). Oleate, palmitate, palmitoleate, linoleate, stearate, cycloheximide, actinomycin D, HepG2, and HEK293T cells were purchased from Sigma-Aldrich (St. Louis, MO, USA). Polyclonal primary antibodies against SCD1 and Actin and secondary antibodies were obtained from Cell Signaling (Danvers, MA, USA). Antibody against Glu-Glu tag was purchased from Bethyl Laboratories (Montgomery, TX, USA). Methanol (gradient grade), isopropanol (gradient grade), n-hexane, and boron trifluoride-methanol solution were purchased from Merck (Rahway, NJ, USA). All other chemicals used in this study were of analytical grade. All experiments and measurements were carried out by using Millipore ultrapure water.

### 4.2. Web-Based Tools for In Silico Analysis

The RNAfold server (http://rna.tbi.univie.ac.at/cgi-bin/RNAfold.cgi, accessed on 24 October 2021) [57] was used to study the stem-loop structure and the stability of the *SCD1* mRNA provided by the SSCprofiler database. This application predicts the secondary structure of a single-stranded RNA and calculates the partition function and base pairing probability matrix as well as the minimum free energy (MFE) structure of the molecule.

The 3D structure of human SCD1 domain was obtained from the PBD protein Data Bank (4ZYO, http://www.pdb.org/pdb/home/home.do; accessed on 9 March 2022) [58]. The 3D structure of the Leu224 variant was generated by the I-TASSER online prediction program (https://zhanggroup.org/I-TASSER/, accessed on 18 October 2021) [59]. All images were rendered using DeepView/Swiss-Pdb Viewer version 4.0.2 (www.expasy.org/spdbv/, accessed on 13 April 2018).

### 4.3. Expression Plasmid Construction and Mutagenesis

*SCD1* cDNA reverse transcribed from HepG2 cell mRNA was cloned into the pcDNA3.1(–) plasmid between the *Xho* I and *Hin*d III restriction sites using 5′–AAA TTT **CTC GAG** CTC AGC CCC CTG GAA AGT GAT–3′ sense and 5′–AAA TTT **AAG CTT** GGA ACC TGA GGG ACC CCA AAC–3′ antisense primers. Bold letters indicate the recognition sites of the two endonucleases, respectively. The studied natural and artificial mutations as well as the Glu-Glu tagged constructs were generated by overlap extension PCR mutagenesis. All constructs were verified by Sanger-sequencing.

### 4.4. Cell Culture and Transfection

Human embryonic kidney (HEK293T) and hepatocellular carcinoma (HepG2) cells were cultured in 12-well plates (5 × 10^5^ cells per well) in Dulbecco’s modified Eagle medium (DMEM) supplemented with 10% fetal bovine serum and 1% penicillin/streptomycin solution at 37 °C in a humidified atmosphere containing 5% CO_2_. HEK293T and HepG2 cells were transfected with 0.5–1 μg pcDNA3.1(–)-SCD1 plasmids using 3 μL Lipofectamine 2000 or 3 μL Lipofectamine 3000 supplemented with 2 µL P3000 (Invitrogen, Carlsbad, CA, USA) in 1 mL DMEM, respectively. Cells were harvested and processed 24–30 h after transfection.

### 4.5. Cell Treatments

For protein stability assay, transfection medium was replaced after overnight incubation at 37 °C with 1 mL DMEM containing the translational inhibitor cycloheximide (50 μg/mL) and the cells were incubated for 1, 2, 4, or 6 h in 12-well plates. Non-transfected samples as well as cells transfected with pcDNA3.1(–) “empty” vector were used as control in all experiments.

For mRNA stability assay, transfection medium was replaced after overnight incubation at 37 °C with 1 mL DMEM containing the mRNA synthesis inhibitor actinomycin D (5 μg/mL) and the cells were incubated for 0, 1, 2, 4, 8, or 12 h in 12-well plates.

Oleate, palmitate, palmitoleate, linoleate, and stearate (Sigma-Aldrich, St. Louis, MO, USA) were diluted in ethanol (Molar Chemicals, Halásztelek, Hungary) to a final concentration of 50 mM, conjugated with 4.16 mM FA free BSA (Sigma-Aldrich, St. Louis, MO, USA) in 1:4 ratio, at 50 °C for 1 h. The working solution for FA treatments was prepared freshly in FBS-free and antibiotic-free medium at 100 µM final concentration. The culture medium had been replaced by FBS-free and antibiotic-free medium 1 h before the cells were treated with FA. The FA-treatment was carried out for 6 h in 12-well plates.

### 4.6. Preparation of Cell Lysates

Cell lysates were prepared for Western blot analysis by removing the medium and washing the cells twice with PBS. 100 μL RIPA lysis buffer (0.1% SDS, 5 mM EDTA, 150 mM NaCl, 50 mM Tris, 1% Tween 20, 1 mM Na_3_VO_4_, 1 mM PMSF, 10 mM benzamidine, 20 mM NaF, 1 mM pNPP, and protease inhibitor cocktail) was added to each well and the cells were scraped and briefly vortexed. After 15 min incubation at room temperature, the lysates were centrifuged for 5 min at maximum speed in a benchtop centrifuge at 4 °C to remove cell debris. Protein concentration of the supernatant was measured with Pierce^®^ BCA Protein Assay Kit (Thermo Fisher Scientific, Waltham, MA, USA) and the samples were stored at −20 °C until downstream analysis.

### 4.7. qPCR Analysis

Total RNA was purified from transfected HEK293T cells by using RNeasy Plus Mini Kit (Qiagen, Germantown, MD, USA) following the manufacturer’s instruction. Possible DNA contamination was removed by DNase I treatment using RNAqueous^®^-4PCR Kit (Invitrogen, Carlsbad, CA, USA). cDNA samples were produced by reverse transcription of 0.5 µg DNA-free RNA using SuperScript III First-Strand Synthesis System for RT-PCR Kit (Thermo Fisher Scientific, Waltham, MA, USA). Quantitative qPCR assay was performed in 20 µL final volume containing 5 µL 20× diluted cDNA, 1× PowerUp^TM^ SYBR^TM^ Green Master Mix, 0.5 µM forward and reverse primers using QuantStudio 12K Flex Real-Time PCR System (Thermo Fisher Scientific, Waltham, MA, USA). Tag-less and Glu-Glu tagged *SCD1* sequences were amplified by 5′–TTG GGA GCC CTG TAT GGG AT–3′, 5′–ACA TCA TTC TGG AAT GCC ATT GTG T–3′ and 5′–CTG GCC TAT GAC CGG AAG AAA–3′, 5′–GAC CCC AAA CTC ATT CCA TAG G–3′ primer pairs, respectively. *GAPDH* cDNA was also amplified as an endogenous control using 5′–GTC CAC TGG CGT CTT CAC CA–3′ and 5′–GTG GCA GTG ATG GCA TGG AC–3′ primers. For increased reliability, RT negative control of each sample was also analyzed in addition to DNase digestion. The first step of the thermocycle was an initial denaturation and enzyme activation at 95 °C for 2 min. It was followed by 40 cycles of 95 °C–15 s, 55 °C–15 s and 72 °C–1 min, measurement of the fluorescent signal was carried out during annealing. Reactions were performed in triplicates, a reaction mixture with RNase-free water instead of template cDNA was employed as non-template control. Relative expression levels were calculated as 2^−Δ*C*_T_^ where Δ*C*_T_ values corresponded to the difference of the *C*_T_-values of the endogenous control and target genes.

### 4.8. Western Blot Analysis

Aliquots of cell lysates (2–5 μg protein per lane) were analyzed by SDS-PAGE on 12% Tris-glycine minigels and transferred onto Immobilon-P membranes (Millipore, Billerica, MA, USA). Primary and secondary antibodies were applied overnight at 4 °C and for 1 h at room temperature, respectively. Horseradish peroxidase (HRP)-conjugated goat polyclonal anti-Actin (Cell Signaling, Danvers, MA, USA, sc-1616) antibodies were used at 1:5000 dilution. SCD1 was detected with a rabbit polyclonal antibody (Cell Signaling, Danvers, MA, USA, 2438S), used at a dilution of 1:2000, followed by HRP conjugated goat polyclonal anti-rabbit IgG (Cell Signaling, Danvers, MA, USA, 7074S) at a dilution of 1:2000. Glu-Glu tag was detected with a goat polyclonal antibody (Bethyl Laboratories, A190-110A), used at a dilution of 1:10,000, followed by HRP conjugated mouse monoclonal anti-goat IgG (Cell Signaling, Danvers, MA, USA, sc-2354) at a dilution of 1:2000. HRP was detected using the SuperSignal West Pico Chemiluminescent Substrate (Thermo Fisher Scientific, Waltham, MA, USA).

### 4.9. GC-FID Analysis of Fatty Acid Profiles

Cells were washed once with PBS, then harvested in 100 µL PBS by scraping for the investigation of the saturated and unsaturated FA content. Samples were centrifuged for 5 min at 1500 rpm in a benchtop centrifuge at room temperature, and supernatants were discarded. Cell pellets were resuspended in 150 µL PBS. 50 µL cell suspension was transferred to a clear crimp vial for GC-FID measurement. 150 µL of methanol containing 2 *W*/*V*% NaOH was added to the 50 µL cell suspension in the crimp vials, the samples were incubated at 90 °C for 30 min, then cooled to room temperature. 400 µL of methanol containing 13–15% of boron trifluoride was added to the samples, and the vials were incubated at 90 °C for 30 min. After cooling to room temperature, 200 µL of saturated NaCl solution and 300 µL of n-hexane were added. Fatty acid methyl esters (FAMEs) were extracted to the upper phase containing n-hexane, and this phase was transferred to a vial for GC analysis. GC analysis was carried out in a Shimadzu GC-2014 gas chromatograph equipped with a Zebron ZB-88 capillary column (60 cm × 0.25 mm i.d., 0.20 µm film thickness) with an 88% (propyl-nitrile)-aryl-polysiloxane stationary phase and a flame ionization detector (FID). For chromatographic separation of FAMEs, the following oven time-temperature program was used: the flow velocity was 35 mL/min, the initial temperature was 100 °C and reached 210 °C with an increase of 5 °C/min. 1 µL of extracted sample was injected into the GC [60].

### 4.10. Subjects

425 patients diagnosed with T2DM in the 2nd Department of Internal Medicine, Semmelweis University (57.7% female, 42.3% male, disease onset at the age of 48.0 ± 12.4 y) were recruited in the study. The control group consisted of 463 volunteers without any medical history of any metabolic disease (58.9% female, 41.1% male, mean age: 39.2 ± 13.0 y). Genetic analysis of the participants was approved by the Local Ethical Committee (ETT-TUKEB ad.328/KO/2005, ad.323-86/2005-1018EKU from the Scientific and Research Ethics Committee of the Medical Research Council). Participants signed written informed consent before sample collection for genetic analysis. To avoid the risk of spurious association caused by population stratification, subjects of Hungarian origin were exclusively included to ensure the comparison of homogenous populations. Buccal epithelial cells were collected by swabs. The first step of DNA isolation was an incubation of the buccal samples at 56 °C overnight in 0.2 mg/mL Proteinase K cell lysis buffer. Subsequently proteins were denatured using saturated NaCl solution. DNA was than precipitated by isopropanol and 70% ethanol. DNA pellet was resuspended in 100 µL 0.5 × TE (1 × TE: 10 mM Tris pH = 8.0; 1 mM EDTA) buffer. Concentration of the samples was measured by NanoDrop1000 spectrophotometer.

### 4.11. Genotyping

The M224L polymorphism (rs2234970) of the *SCD1* gene was genotyped by own-designed TaqMan assay. The assay contained the two primers (sense: 5′–CAC AAG CGT GGG CAG GAT–3′, antisense: 5′–GGT GTC TGG TCT GTC AAT GTA GGT–3′) and the allele-specific fluorescent probe labelled by JOE for the C allele (5′–***JOE***-AAG CAC ATC A**G**C AGC AAG CCA GGT T-***BHQ1***–3′) and labelled by FAM for the A allele (5′–***FAM***-AAG CAC ATC A**T**C AGC AAG CCA GG-***BHQ1***–3′). Real-time PCR assay was performed in 10 µL final volume containing approximately 4 ng genomic DNA, 1× TaqPath™ ProAmp™ Master Mix, 0.5 µM primers and 0.3 µM allele-specific probes using QuantStudio 12K Flex Real-Time PCR System (Thermo Fisher Scientific, Waltham, MA, USA). Thermocycle was started by activating the hot start DNA-polymerase and denaturing genomic DNA at 95 °C for 10 min. This was followed by 40 cycles of denaturation at 95 °C for 15 s, and combined annealing and extension at 60 °C for 1 min. Real-time detection was carried out during the latter step to verify the results of the subsequent post-PCR plate read and automatic genotype call.

### 4.12. Statistics

The results of the Western blot analyses were evaluated by densitometry using ImageQuant 5.2 software and are shown as relative band densities normalized to Actin as a reference. Data are presented in the diagrams as mean values ± S.D. and were compared by ANOVA with Tukey’s multiple comparison post hoc test using GraphPad Prism 6 software. Differences with a *p* < 0.05 value were considered to be statistically significant. Genotype–phenotype association was assessed by *χ*^2^-test.

## Figures and Tables

**Figure 1 ijms-23-06221-f001:**
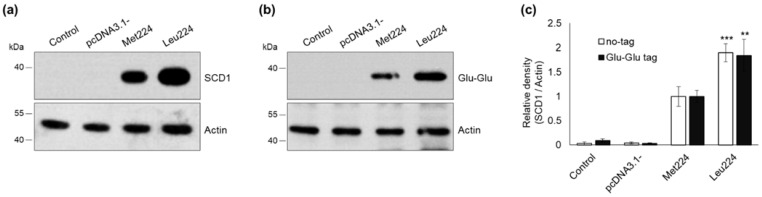
Expression of M224L variants of stearoyl-CoA desaturase-1 in transiently transfected HEK293T cells. HEK293T cells were harvested and processed 24 h after transfection. Aliquots of cell lysates (5 µg) were loaded on 12% SDS-polyacrylamide gel, transferred to Immobilon-P membrane and SCD1 was detected with an anti-SCD1 (**a**) and an anti-Glu-Glu tag (**b**) antibodies, respectively. Actin was measured as loading control. Representative immunoblots of three independent experiments are shown. The band intensities were quantitated by densitometry and SCD1/Actin ratios are shown as bar graphs (**c**). Data are shown as mean values ± S.D. Statistical analysis was performed with the Tukey–Kramer Multiple Comparisons Test. ** *p* < 0.01; *** *p* < 0.001.

**Figure 2 ijms-23-06221-f002:**
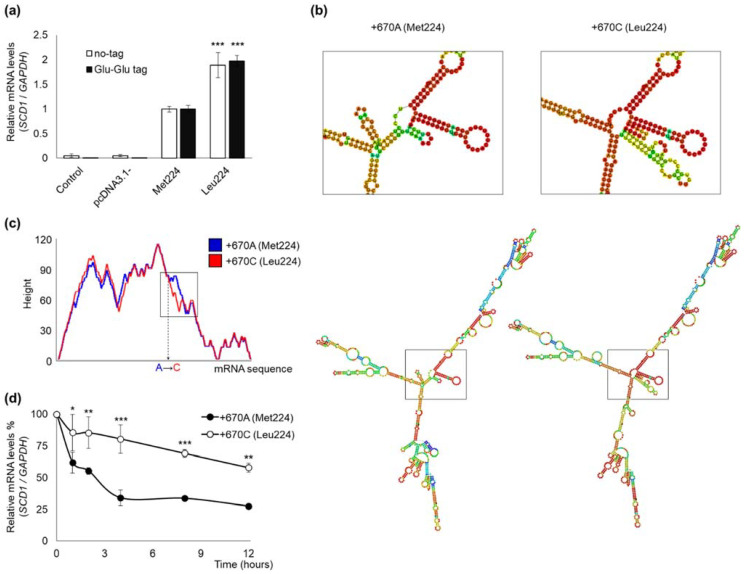
mRNA levels and structures of rs2234970 polymorphic *SCD1* variants. (**a**) The mRNA level was measured in transiently transfected HEK293T cells. Samples were harvested and prepared as described in Section 4. qPCR was carried out using *GAPDH*, *SCD1* and Glu-Glu tag sequence specific primers as indicated in Section 4. The diagram presented depicts the results of three independent measurements. Statistical analysis was performed with the Tukey–Kramer Multiple Comparisons Test. Data are shown as mean values ± S.D. *** *p* < 0.001. (**b**) The secondary MFE structures of +670A and +670C *SCD1* were predicted by RNAfold server as indicated in Section 4. mRNA numbering begins at the first nucleotide of the start codon. The difference in the secondary structure of mRNAs is magnified in the left corner of the figure. (**c**) Mountain plot representation of MFE structure. The mountain plot represents a secondary structure in a plot of height vs. position, where the height is given by the number of base pairs enclosing the base at given position. Loops corresponding to plateaus, hairpin loops to peaks, and helices to slopes. The position of the polymorphic nucleotide change is indicated on horizontal axis. Red line: +670A; blue line: +670C. (**d**) HEK293T cells were transiently transfected with Glu-Glu tagged Met224 and Leu224 expressing constructs. 24 h after transfection cells were treated with actinomycin D (5 µg/mL) for 0, 1, 2, 4, 8, or 12 h. Samples were harvested and prepared as described in Section 4. qPCR was carried out using *GAPDH* and Glu-Glu tag sequence specific primers as indicated in Section 4. The diagram presents the average of three parallels. Statistical analysis was performed with the Tukey–Kramer Multiple Comparisons Test. Data are shown as mean values ± S.D. * *p* < 0.05; ** *p* < 0.01; *** *p* < 0.001.

**Figure 3 ijms-23-06221-f003:**
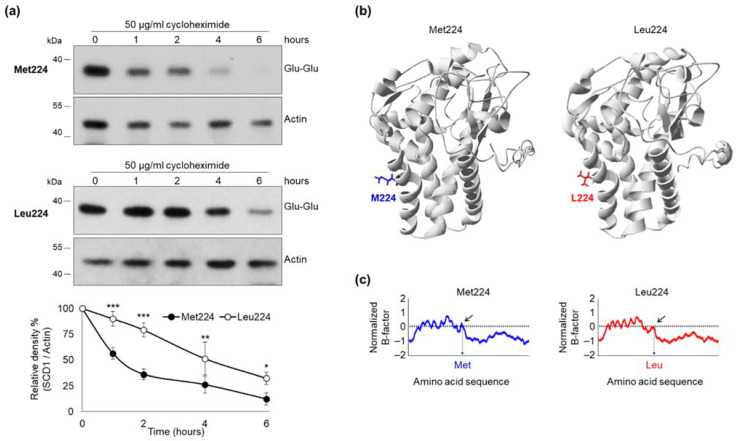
Intracellular degradation of M224L variants. (**a**) HEK293T cells were transiently transfected with Glu-Glu tagged Met224 and Leu224 expressing constructs. 24 h after transfection cells were treated with cycloheximide (50 mg/mL) for 0, 1, 2, 4, and 6 h. Western blot analysis of cell lysates (2 µg protein per lane) was carried out using anti-Glu-Glu tag and anti-Actin antibodies. Representative result of four independent experiments is shown. Data are shown as mean values ± S.D. The band intensities were quantitated by densitometry and SCD1-Glu-Glu/Actin ratios are shown as bar graphs. Statistical analysis was performed with the Tukey–Kramer Multiple Comparisons Test. Data are shown as mean values ± S.D. * *p* < 0.05; ** *p* < 0.01; *** *p* < 0.001. (**b**) Resolved crystal structure of Met224 human SCD1 was obtained from Protein Data Bank (file 4ZYO). 3D structure of Leu224 polymorphic variant was predicted by I-TASSER. Images were rendered using DeepView/Swiss-Pdb Viewer version 4.0.2. Met224 is shown in blue, Leu224 is shown in red. (**c**) Comparison of normalized B-factor values for Met224 and Leu224 SCD1 variants. The normalized-B factor (B-factor of each residue/B-factor of whole average) was plotted as the function of the amino acid residues. The blue line indicates the Met224 protein, and the red line shows Leu224. The position of the polymorphic residue change is indicated on the horizontal axis.

**Figure 4 ijms-23-06221-f004:**
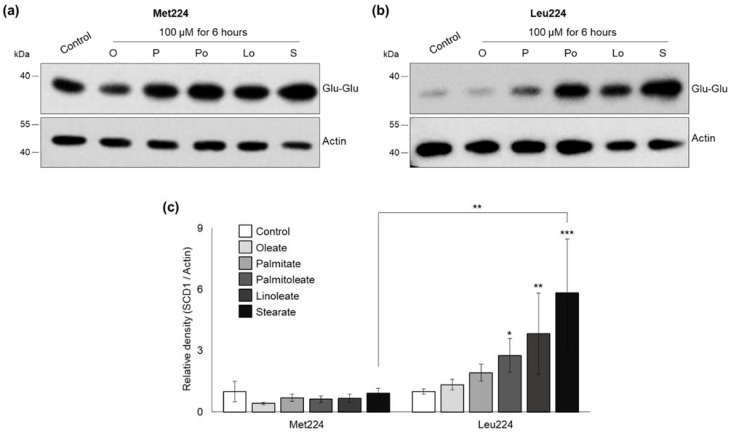
Effect of various fatty acids on the expression of M224L SCD1 variants in HEK293T cells. 24 h after transient transfection Met224 (**a**) and Leu224 (**b**) expressing cells were treated with BSA-conjugated oleate, palmitate, palmitoleate, linoleate, and stearate to a final concentration of 100 μM for 6 h. Immunoblot analysis of cell lysates (2 µg protein per lane) was carried out using anti-Glu-Glu tag and anti-Actin antibodies. Representative result of five independent experiments is shown. (**c**) The band intensities were determined by densitometry and the SCD1-Glu-Glu/Actin ratios were plotted. Statistical analysis was performed with the Tukey–Kramer Multiple Comparisons Test. Data are shown as mean values ± S.D. * *p* < 0.05; ** *p* < 0.01; *** *p* < 0.001.

**Figure 5 ijms-23-06221-f005:**
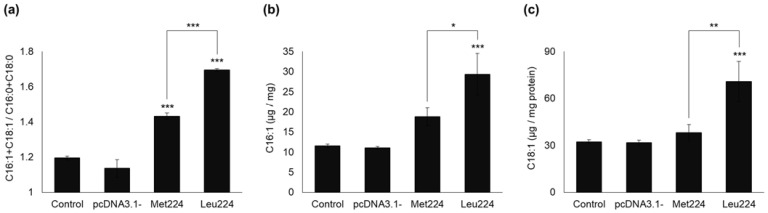
Changes in cellular fatty acid desaturation upon the over-expression of M224L variants. Transiently transfected HEK293T cells were harvested and the amount of major saturated (C18:0, C16:0) and mono-unsaturated ((**a**) C18:1 cisΔ9; (**b**) C16:1 cisΔ9) fatty acids was measured by GC-FID after saponification and methylation as described in Section 4, (**c**) and the ratio of unsaturated: saturated fatty acids was calculated. Data were normalized to the total protein content of the samples and are shown as mean values ± S.D.; *n* = 3. Statistical analysis was performed with the Tukey–Kramer Multiple Comparisons Test. * *p* < 0.05; ** *p* < 0.01; *** *p* < 0.001.

**Figure 6 ijms-23-06221-f006:**
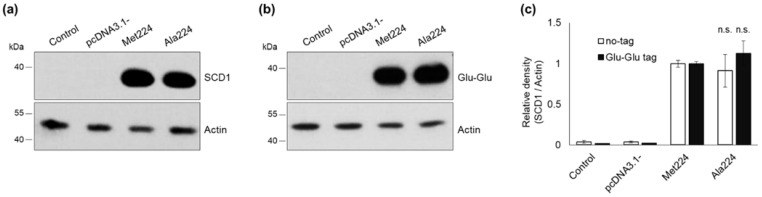
Intracellular protein level of Ala224 artificial SCD1 variant. HEK293T cells were harvested and processed 24 h after transfection. Aliquotes of cell lysates (5 µg) were loaded on 12% SDS-polyacrylamide gel, transferred to Immobilon-P membrane and SCD1 was detected with an anti-SCD1 (**a**) and an anti-Glu-Glu tag (**b**) antibody, respectively. Actin was measured as loading control. Representative immunoblots of three independent experiments are shown. The band intensities were quantitated by densitometry and SCD1/Actin ratios are shown as bar graphs (**c**). Data are shown as mean values ± S.D. Statistical analysis was performed with the Tukey–Kramer Multiple Comparisons Test. n.s.: non-significant difference 4 h after the translation arrest, the intracellular amount of Met224 SCD1 was reduced to approximately 25%, while about 50% of the Leu224 and Ala224 proteins could still be detected at the same time (see Figure 3a and Figure 7a). Unlike the Leu224 version, however, the Ala224 protein level was not affected by oleate, palmitate, palmitoleate, linoleate, or stearate (Figure 7b). Also, to the contrary of Leu224 *SCD1*, the mRNA level of Ala224 *SCD1* was significantly, by about 30%, lower than that of the Met224 variant (Figure 7c).

**Figure 7 ijms-23-06221-f007:**
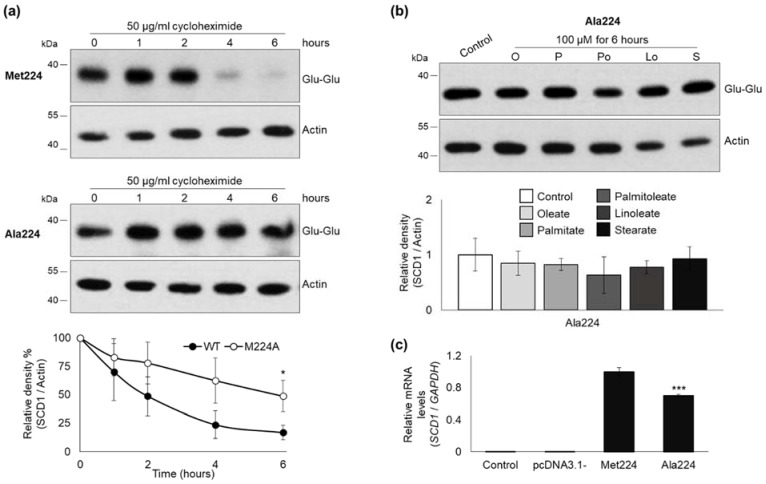
Expression, stability, and fatty acid sensitivity of Ala224 artificial SCD1 variant. (**a**) HEK293T cells were transiently transfected with Glu-Glu tagged Met224 and Ala224 constructs. 24 h after transfection cells were treated with cycloheximide (50 mg/mL) for 0, 1, 2, 4, and 6 h. Western blot analysis of cell lysates (2 µg protein per lane) was carried out using anti-Glu-Glu tag and anti-Actin antibodies. Representative result of four independent experiments is shown. The band intensities were determined by densitometry and the SCD1-Glu-Glu/Actin ratios were plotted. (**b**) 24 h after transient transfection, cells were treated with BSA-conjugated oleate, palmitate, palmitoleate, linoleate, and stearate to a final concentration of 100 μM for 6 h. Immunoblot analysis of cell lysates (2 µg protein per lane) was carried out using anti-Glu-Glu tag and anti-Actin antibodies. Representative result of three independent experiments is shown. The band intensities were quantitated by densitometry and SCD1-Glu-Glu/Actin ratios are shown as bar graphs. (**c**) qPCR was elaborated using *GAPDH* and *SCD1*-Glu-Glu tag sequence specific primers as indicated in Section 4. The diagram presented depicts the results of three independent measurements. Statistical analysis was performed with the Tukey–Kramer Multiple Comparisons Test. Data are shown as mean values ± S.D. * *p* < 0.05; *** *p* < 0.001.

**Figure 8 ijms-23-06221-f008:**
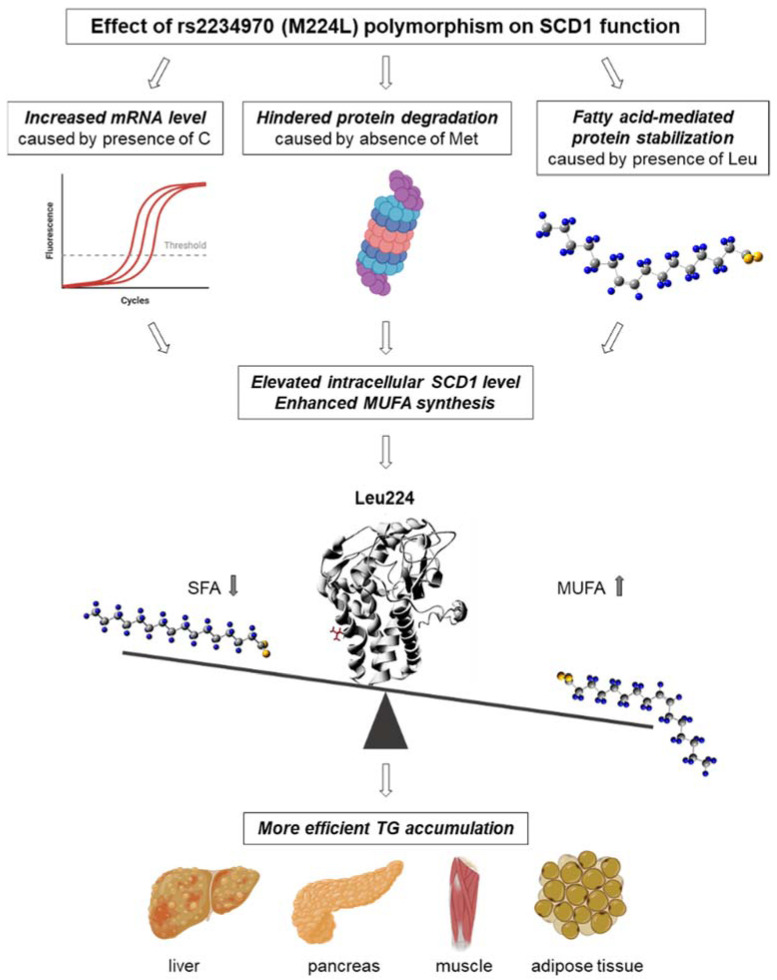
Potential health impact of rs2234970 (M224L) SCD1 polymorphism and its proposed molecular background.

**Table 1 ijms-23-06221-t001:** Allele-, genotype-, and genotype combination frequencies in control and T2DM groups.

		Control (N = 463)		T2DM (N = 425)	
		N	%	N	%
**Allele**	**Met**	574	62	502	59
	**Leu**	352	38	348	41
	** *χ^2^* **	*p* = 0.2071			
**Genotype**	**Met/Met**	181	39	152	36
	**Met/Leu**	212	46	198	47
	**Leu/Leu**	70	15	75	18
	** *χ^2^* **	*p* = 0.4601			
**Genotype combination**	**−Leu**	181	39	152	36
	**+Leu**	282	61	273	64
	** *χ^2^* **	*p* = 0.3061			

## Data Availability

Data is contained within the article.

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
