# Peer review of "Molecular Mechanisms Underlying the Elevated Expression of a Potentially Type 2 Diabetes Mellitus Associated SCD1 Variant"

_ijms, 2022, doi:10.3390/ijms23116221_

Round 1
Reviewer 1 Report
IJMS-1733470-Review
Dysregulation of lipid metabolism is known to be a major risk of various metabolic disorders. In this regard, Stearoyl-CoA desaturase-1 (SCD1) has an essential role in these diseases, as it catalyzes the synthesis of unsaturated fatty acids, for fat storage and contributing to cellular defense against saturated fatty acid toxicity. Recent studies showed that increased activity or overexpression of SCD1 is one of the contributing factors for type 2 diabetes mellitus (T2DM). In this study, the authors aimed to investigate the impact of the common missense rs2234970 (M224L) polymorphism on SCD1 function in transfected cells. The authors found that the Leu224 variant results in higher intracellular enzyme levels due to a combination of (i) increased mRNA stability, (ii) decreased protein degradation rate and (iii) susceptibility to protein stabilizing effects of different FAs. Moreover, they demonstrated that expression of the Leu224 variant increased the cellular unsaturated : saturated FA ratio more effectively than that of the Met224 version of the enzyme. In addition, a mild, yet statistically not significant accumulation of the Leu224 allele was also found in a group of T2DM patients.
This is an interesting and important research. The authors performed also experiments using complementary methods trying to support their data; however, there are some major concerns that should be addressed.
Specific comments
- It is appreciated that expression levels of SCD1 protein was determined in two cancer cell lines, HEK293T and HepG2; however, as results are very similar Figure 1d-f should be moved to the supplementary material and Figure 1a-c and Figure 2 should be combined to one Figure demonstrating expression levels at both protein and mRNA levels. In the text, these two results should also be combined under a combined subtitle (combine 2.1 and 2.2 subheadings).
In addition, stability of mRNA can be measured directly; as in the case of protein synthesis inhibitor, using an mRNA synthesis inhibitor such as a-amanitin. This should be performed and added to the results to support mRNA stability of one of the variants yes/no.
- Figure 3a – It seems that there is a serious problem regarding fig. 3a; the upper and lower blots are actually the same blot exposed for different time-periods and don’t really represents the two variants. The authors should repeat the experiment, running the two variants (or represent another blot as they claim for running it in four independent experiments) to confirm indeed protein stability.
-Figure 4b-The density graph of the two variants should be plotted with two different scales (SCD1/Actin); one for the Met224 and one for the Leu224; in such a case it will be obvious that the Met224 variant is also affected by the various FAs; however, may be to a less extant. The Leu224 blot is over-exposed and thus hard to measure the density; authors should represent a less exposed blot for the Leu224. Upon the new represented data, authors should correct the text and their conclusions.
Figure 4c-The authors checked the effect of the FAs on the mRNA levels of the two variants and concluded that none of the FAs alter the mRNA levels. However, in the design of the experiment, 100uM of each FA added for 6 hr; one can’t expect any effect of FAs on the synthesis at the mRNA levels; this should be checked upon longer exposure of the cells to FAs. Thus, it is recommended that fig. 4c shouldn’t be represented at all or moved the supplementary material. If the authors would like to state any conclusion regarding the effect of FAs on the two variants at the mRNA levels; they should repeat the experiments performed for longer FAs treatment periods.
-Legend to Figure 7; please exchange between b and c in the explanation.
-Table 1: although numbers of control and T2DM patients include a few hundreds and are similar; none of the result is significant; thus no conclusion can be stated regarding the frequency of the leucine allele in T2DM patients. Thus, also discussion should be less oriented to the connection made between the SCD1 polymorphism and T2DT and rewritten accordingly; including change in figure 8 (deleting the part of the risk for T2DM).
In this regard, a major drawback of this manuscript is in its definitive conclusions that do not really reflect the obtained results.
Author Response
Please see the attachement.

Reviewer 2 Report
The paper entitled “Molecular mechanisms underlying the elevated expression of a potentially type 2 diabetes mellitus associated SCD1 variant” includes potentially relevant data for the biochemistry of liver, pancreas, muscles and adipose tissue as well as for the pathobiochemistry of metabolic disorders associated with the overweight/obesity. The Authors of this publication tend to suggest that the minor variant of rs2234970 polymorphism might contribute to the development of obesity-related metabolic disorders including type 2 diabetes mellitus through an increased intracellular level of Stearoyl-CoA desaturase-1.
Remarks:
- The Authors of the work in the title [Molecular mechanisms underlying the elevated expression of a potentially type 2 diabetes mellitus associated SCD1 variant] of the publication refer to common metabolic disorder. Therefore, it should be mentioned – at least briefly – about type 2 diabetes, in the Introduction Subsection.
- According to the part of Figure 3. (a) concerning the “Relative density % (SCD1 / Actin)/Time (hours)” – no information on statistical significance in the above-mentioned graph can be found – the lack of relevant information should be supplemented/explained
- According to the Figure 4. (c) concerning the “Relative mRNA levels (SCD1 / GAPDH)/Met224/Leu224” – no information on statistical significance in the above-mentioned graph can be found – the lack of relevant information should be supplemented/explained
- According to the Figure 6. (c) concerning the “Relative density % (SCD1 / Actin)/ Control/pcDNA3.1-/Met224/Ala224” – no information on statistical significance in the above-mentioned graph can be found – the lack of relevant information should be supplemented/explained
- Selected parts of the Discussion section [e.g. lines: 311-317; lines: 355-358; lines: 359-365; lines: 366-373; lines: 374-383] have not been referenced and have not been compared with the current literature data – it need to be completed.
- Summary subsection should be modified – the mentioned part of the article – important, if not the most important, describes the results. In the subsection summary - as well as throughout the text - it is difficult to find information on the potential applications of the results described by the Authors – in the fields of the medicine or molecular diagnostics of metabolic diseases.
Round 2
Reviewer 1 Report
Authors accpted most of the suggestions and corrected the manuscript accordingly; which greatly improved the scientific merit of the paper.
Reviewer 2 Report
The Authors of the publication entitled: “Molecular mechanisms underlying the elevated expression of a potentially type 2 diabetes mellitus associated SCD1 variant” responded to the comments of the reviewer.
The manuscript has been modified so that it is suitable for publication in its present shape.